# The Influence of Pretreatment on the Efficiency of Electrochemical Processes in Oily Wastewater Treatment

Morana Druskovic [1],*, Drazen Vouk [2], Tomislav Bolanca [3] and Hana Posavcic [2]

1 Indeloop Ltd., 10000 Zagreb, Croatia
2 Water Research Department, Faculty of Civil Engineering, University of Zagreb, 10000 Zagreb, Croatia
3 Department of Analytical Chemistry, Faculty of Chemical Engineering and Technology, University of Zagreb, 10000 Zagreb, Croatia
* Correspondence: morana.druskovic@dok-ing.hr; Tel.: +385-98-914-4736

**Abstract:** Wastewater containing oil is becoming a growing problem worldwide due to increasing quantities and existing pollution. The pollutants contained in these effluents, when released into the environment, affect surface and groundwater pollution, endanger human life and health, and pollute the atmosphere. Their sustainable treatment should be cost-effective and meet all requirements to prevent the pollutants from being transferred to the environment or to humans. This study gives a brief overview of some conventional and modern technologies that have been proven in practice for the treatment of oily wastewater. Due to the high concentrations of chemical oxygen demand (COD) and total hydrocarbons (mineral oils) in oily wastewater its treatment is complex, and to achieve optimum treatment conditions and efficiency a combination of different technologies is required. This paper focuses on hybrid electrochemical process combining the electro-Fenton process (EF) using stainless steel (SS), and electrocoagulation (EC) with iron (Fe) and aluminum (Al) electrodes. The influence of the two different types of pretreatment, i.e., pretreatment of the raw wastewater on the overall efficiency of oily wastewater treatment using a hybrid treatment process, which is a combination of AOP and EC, is investigated. Two type of pretreatment were tested, with primary sedimentation and pretreatment of the mixture of raw wastewater and previously generated electrochemical sludge with primary sedimentation. During the applied treatment processes, the concentration of COD, mineral oils, and other elements in the raw and treated wastewater (As, Ca, Cd, Cr, Cu, Ni, Pb, Sn, Zn) and in the generated sludge (K, Ca, Fe, Ti, V, Cr, Mn, Ni, Cu, Zn, Ga, As, Br, Rb, Sr, Y, Zr, Pb, Th) were determined. By combining the primary sedimentation of the raw wastewater with the EF/EC process, a mineral oil removal efficiency of 72% (1.1 mg/L) and COD of 89% (170 mg/L) was achieved. Using primary sedimentation of a mixture of raw wastewater with previously generated sludge as pretreatment and followed by EF/EC treatment, a higher efficiency for mineral oils of 94% (7.6 mg/L) and COD of 98% (43 mg/L) was achieved.

**Keywords:** oily wastewater; treatment; primary sedimentation; electrochemical processes; electro-Fenton; electrocoagulation; electrochemical sludge; mineral oil; COD

## 1. Introduction

As urbanization and industrialization increase, so does the amount of wastewater that must be treated before it can be discharged into the environment or reused. When these effluents are discharged without prior treatment, there may be an increase in total suspended solids (TSS), biochemical oxygen demand (BOD), chemical oxygen demand (COD), oil, grease, etc., which leads to a disturbance of the aquatic ecosystem and might endanger human health [1]. The development of industry leads to an increase in the amount of waste oils, and most of them are generated in petrochemical, metallurgical, mechanical, and marine industries [2]. Oily wastewater is defined as a combination of wastewater and oil in a certain ratio [3]. The wastewater generated by the aforementioned

industries may contain toxic chemicals, hydrocarbons, heavy metals, microorganisms, biological substances, microplastics, oils, and viruses, and are carcinogenic and mutagenic to human health [1].

Many countries set limits on the maximum allowable concentration of oil discharges in the range of 5–100 mg/L [1]. In some countries of the world, the limits for wastewater emissions discharged into the environment or public sewage system are prescribed by law and are based on the allowable concentrations of pollutants and/or contaminants in the wastewater. Industrial wastewater, where most oil and grease are generated before being discharged into the public sewer system, must be pre-treated in some form according to EU regulations [3,4].

Oil-containing wastewater is usually in emulsified form, and this form differs from a dispersed solution mainly in its stability. The improved stability is influenced by substances-emulsifiers-located at the boundary between oil and water [3]. Based on their physical properties, oils are divided into four categories-free, dispersed, emulsified and dissolved oils. Free floating oil particles can be removed mechanically, whereas unstable oil emulsions are broken chemically and separated by gravity [2,5]. The most commonly used conventional technologies for treating oily wastewater (Table 1) can be divided into four main categories: physical, mechanical, chemical, and biological. Conventional processes for treating oily wastewater often do not ensure satisfactory treatment performance when applied independently, and treatment plants based on these technologies are characterized by increased costs for construction, operation, and maintenance, which makes them less acceptable in practice [3]. An overview of some conventional technologies with their advantages and disadvantages can be found in Table 1 [1,4,6–21].

**Table 1.** Advantages and disadvantages of conventional methods and modern technologies for the treatment of oily wastewater.

| Methods | Advantages | Disadvantages |
|---|---|---|
| | Conventional | |
| Gravity separation (GS) | System is very simple | Cannot be used to separate emulsified oil |
| Flocculation | Easier process management, lower capital and maintenance costs | Large volume of sludge generation |
| Demulsifiers | Processing time less that 30 min | Long duration of microbial cultivation and the unstable effect of demulsification |
| Chemical emulsion breaking (CEB) | Activated carbon has very good adsorption properties | Expensive and complicated regeneration, unstable to changes in pH, salinity of water, exposure time and temperature |
| Microwave irradiation | No chemicals are used and secondary pollution | High setup and installation cost |
| Mechanical coalescers (MC) | Do not require much space for the construction, have compact structure and long service life | Suitable for offshore, complex design of coalescer |
| | Modern Technologies | |
| Biological treatment (BT) | Effective in treating wastewater with high temperatures and high pollutant concentrations | Unstable and their effectiveness is limited for toxic chemicals and water with high salinity |
| Aerobic granular activated sludge reactor (AGR) | Good stability and sedimentation characteristics, stability against toxic pollutants, does not require large volumes | |
| Membrane bioreactor (MBR) and sequencing batch bioreactor (SBR) | Effective in removing large concentrations of organic load and hydrocarbons | Low stabilitiy, membrane fouling, alteration of biokinetics and high salinity |
| Membrane separation technologies (MST) | Low maintenance costs, and energy efficiency | Low mechanical and chemical resistance, membrane fouling, not suitable for heavily fouled oily wastewater |
| Polymeric membranes (PM) | Oily wastewater with low organic matter load | Low mechanical strength, thermal stability, and chemical resistance |
| Ceramic membranes (CM) | Higher porosity, defined pore distribution, higher flow at lower pressures, better separation efficiency, and good chemical, thermal, and mechanical stability | Difficult to handle because they are very fragile and have high manufacturing costs |

Conventional methods have shown many limitations, so some modern techniques have been developed that have proven effective in removing oil from oily wastewater. The best known are biological treatment (BT), supercritical water oxidation (SCWO), microelectrolysis, and membrane separation technology (MST). The future directions of technology development are mainly focused on the research and development of the BT and MST to achieve the highest efficiency in oil removal with minimum cost of construction, operation and maintenance, and to overcome the main drawbacks of these processes such as high salinity of wastewater and high input pollution [1,22–36].

## 2. Electrochemical and Advance Oxidation Processes (AOP)

In the last decade, research based on the application of advanced electrochemical methods in the treatment of drinking water and wastewater has been increasingly intensified. The research is concerned with the treatment of different types of wastewater: domestic wastewater, industrial wastewater, water contaminated with heavy metals, organic and inorganic pollutants, dyes, underground contaminated water, leachate, etc. A more recent approach to treatment involves the use of electrochemical (electrocoagulation, EC) and advanced oxidation processes (electro-Fenton, EF). For each wastewater, it is observed how electrochemical processes alone or in combination with AOP affect its composition and what percentage of pollution can be removed to make the water safe for use or discharge to the environment [3]. Both methods have become the subject of numerous studies because they treat wastewater using electricity instead of chemical reagents and the biological activity of microorganisms, which may be less economical and operationally beneficial in certain circumstances. Previous studies have shown that further research is needed in the field of combining electrochemical and AOP, i.e., electrocoagulation and electro-Fenton processes.

Electrochemical methods involve applying an electric field to one or more sets of electrodes immersed in oily wastewater, with or without the use of semipermeable membranes or additional electrolytes, to remove inorganic, organic, and microbiological contaminants from the water. Depending on the concept of the system, a distinction is made between electrocoagulation, electroflotation, electrooxidation, and electrodialysis. Electrochemical processes are not distinguished by the treatment mechanism, but by the fact that the substances necessary for carrying out the process are generated in situ in a reaction vessel designed as an electrochemical cell. Under the influence of the electric field of the sacrificial anode, the cations (e.g., $Fe^{2+}$, $Al^{3+}$) necessary for the process of coagulation of the pollutants present in the water are released with simultaneous oxidation of the water into $O_2$ and $H^+$ ions. At the same time, the water is reduced at the cathode, producing $H_2$ and $OH^-$ ions. The reaction of cations and $OH^-$ ions leads to the formation of stable hydroxides and polyhydroxides of iron and aluminum. Polyhydroxides have a large surface area and act as flocculants for adsorption of emulsified oil in wastewater. In addition, the $H_2$ and $O_2$ gas microbubbles generated in the system can adsorb oil droplets and assist their removal to the surface of the water column, i.e., flotation. The interaction of bubbles and oil droplets occurs in four steps. The first step is the collision of bubbles and droplets, where they flow together, followed by the formation of agglomerates, which form flocs. The final step is the cleaning of the water column with flocs floating to the surface. Suspended and dissolved impurities are removed by coagulation with electrochemically generated iron and aluminum cations, co-precipitation with iron and aluminum hydroxides, and precipitation of corresponding metal hydroxides [1,3,37].

To achieve the best efficiency of the process at the lowest possible treatment cost, process parameters such as electrode material, distance between electrodes, duration of reactions at each electrode, current intensity, electrical conductivity, and pH values must be optimized. Some of the main advantages of this process are energy efficiency and the ability to adapt to larger and smaller capacities. Previous studies have shown that the efficiency of oil removal by the electrocoagulation method is about 99% [1].

AOPs are processes in which highly reactive radicals are generated in sufficient concentrations under the influence of energy, chemistry, electricity, or radiation to degrade organic

compounds in wastewater. Some of the oxidizing species generated in these processes are superoxide radicals ($O_2^-$) and hydroperoxyl radicals ($HO_2^-$), and the best known are hydroxyl radicals ($OH^-$) [38,39]. The oxidation of organic material is carried out by indirect anodic oxidation, in which the oxidation of organic material is carried out by electrochemically generated reactive oxygen species (chlorine, hypochlorite, hydrogen peroxide, ozone) [40].

Electrochemical advanced oxidation processes (ENOP) belong to the group of AOP, of which the best known are electrolytic oxidation or anodic oxidation (AO) and indirect anodic oxidation or electro-Fenton process (EF). The mechanism of AO is based on the direct generation of hydroxyl radicals ($OH^-$), whereas in EF, there is an indirect catalytic generation of $OH^-$ radicals in the solution to be treated. In the EF process, the Fenton reagent, an oxidative mixture of hydrogen peroxide ($H_2O_2$) and divalent iron ($Fe^{2+}$) as catalyst, is electrochemically generated at the cathode. $H_2O_2$ is generated by the reduction of dissolved $O_2$ in an electrochemical cell at pH $\approx$ 3, and the $Fe^{2+}$ ion is generated by the reduction of $Fe^{3+}$. The electrocatalytic regeneration of $Fe^{2+}$ ions to $Fe^{3+}$ allows $Fe^{3+}$ ions to be reused as a catalyst, avoiding the formation of large amounts of iron hydroxide sludge. When iron electrodes are used, indirect anodic oxidation of organic material takes place with $OH^-$ radicals generated from electrochemically produced $Fe^{2+}$ and $H_2O_2$ [40–43].

In his research, Zhang (2017) used a combination of microelectrolysis, Fenton oxidation, and coagulation to treat wastewater from oilfield fracturing. Microelectrolysis and Fenton oxidation were used to remove the organic load, expressed as COD. The process of microelectrolysis results in the breakup of long carbon chains, oxidation, and redox electrocoagulation of the organic compounds present in the wastewater. The optimal conditions for this process were determined: pH 3, iron-carbon 50 mg/L, mass ratio iron-carbon 2:3, and reaction time 60 min. To achieve optimal Fenton oxidation, the optimal time is 90 min, $H_2O_2$ 12 mg/L, molar ratio 30 $H_2O_2$/$Fe^{2+}$, pH 3, whereas the optimal conditions for the coagulation process are pH 4.3, polyacrylamide 2 mg/L, and mixing speed 150 rpm and 30 s. By combining these processes, an overall efficiency of 85.23% can be achieved, with the contribution of the microelectrolysis process being 68.45%, Fenton oxidation 24.07%, and coagulation 7.48% [44].

In their study, Ahmed et al. (2021) investigated the effects of the EF process on oil refinery wastewater at different pH values. In this study, iron and stainless steel electrodes connected to a DC power source were used with a stirring speed of 250 rpm, a voltage of 29.6 V, and with the addition of 0.2 g NaCl. The efficiency of organic load removal decreased as the pH increased from 2 to 9. In this study, the optimal conditions were determined to be pH 3, current 1 A, 35 mg/L $H_2O_2$, and a reaction time of 30 min to achieve a maximum organic load removal efficiency of 98% [45].

The main objective of the research is to investigate the influence of the type of pretreatment on the efficiency of treating oily wastewater in combination with the EF/EC process. In this study, two types of pretreatment were selected: primary sedimentation of raw wastewater and primary sedimentation of the mixture of raw wastewater and previously generated sludge. The concentration of organic load was monitored with two parameters, mineral oil and COD. In addition, the concentrations of some inorganic indicators in the wastewater and sludge were monitored, which are generated during the treatment process due to the material of the electrodes used in the treatment.

## 3. Materials and Methods

### 3.1. Oily Wastewater Sampling

Effluents were obtained from oil and grease separators from traffic areas in the Republic of Croatia. Before each experiment, the effluents were first thoroughly mixed to obtain a homogeneous mixture (600 rpm/10 min). Before each experiment, homogeneous samples of the raw wastewater were analyzed to distinguish the initial properties of the water.

*3.2. Treatment of Oily Wastewater*

The collection of a sample of the raw wastewater for analysis was followed by the pretreatment of the oily water. All experiments were performed batchwise in the pilot plant shown in Figure 1 at a room temperature of 22 °C. The pilot plant consists of a main reactor and a separator. The effectiveness of two types of pretreatment was tested-primary sedimentation of raw wastewater and primary sedimentation of a mixture of raw wastewater with previously generated sludge. Pretreatment was used to reduce the load of contaminated wastewater to the main electrochemical reactor and to achieve higher final treatment efficiency.

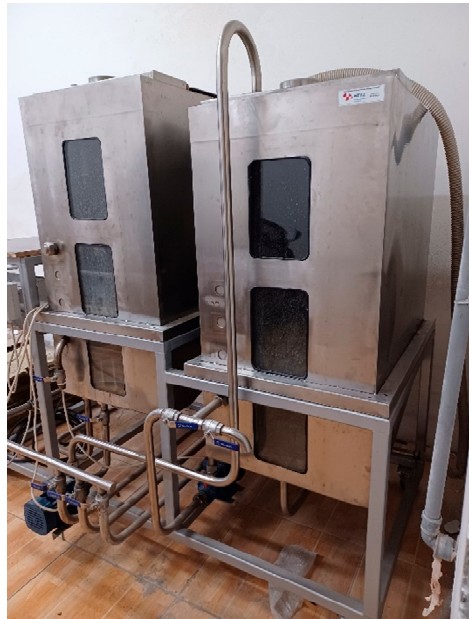

**Figure 1.** Pilot plant for the treatment of oily wastewater using pretreatment and EF/EC processes.

3.2.1. Primary Sedimentation of Raw Wastewater and EF/EC Process

The efficiency of pretreatment of oily wastewater in primary sedimentation of raw wastewater was tested. The raw wastewater was first mixed to obtain a homogeneous suspension, then allowed to settle by gravity for 24 h. Then, it was decanted and 80 L of wastewater was collected.

Pretreatment of the oily wastewater was followed by the main treatment EF/EC. The main treatment was carried out in batches. The first step includes treatment with a set of stainless steel electrodes (20 min, 10 A), in which oxidation of organic matter is carried out with electrochemically generated reactive oxygen species (chlorine, hypochlorite, hydrogen peroxide, ozone). This was followed by settling and decantation. Then followed electroreduction with a set of iron electrodes (15 min, 10 A), with precipitation and decantation. The final step involved coagulation with a set of aluminum electrodes (20 min, 10 A), settling, and decantation. The set of stainless steel electrodes consisted of 12 rectangular plates (400 × 40 mm), and the set of iron and aluminum electrodes consisted of 6 parallel stacked plates with the same dimensions as the stainless steel plates and active surface of 146.7 cm$^2$. The thickness of the stainless steel plates is 2 mm, iron plates 3 mm, and aluminum plates 7 mm. All plates are separated from each other by a 5 mm thick insulator. The electrodes on both sides represent the active surface, each odd electrode is the cathode, and the even is the sacrificial anode. During EF/EC wastewater treatment, the suspension was mixed with air using a TetraTec APS 400 pump. All experiments were performed in batches in a pilot plant (Figure 1). Each set of electrodes was placed on the bottom of the reactor vessel and connected to a Mean Well RSP-3000-12 laboratory rectifier.

### 3.2.2. Primary Sedimentation of Mixture of Raw Wastewater with Previously Generated Electrochemical Sludge and EF/EC Process

Electrochemical sludge is produced during the applied EF/EC process of oily wastewater treatment. After the treatment process, the wastewater was allowed to settle and then the treated water was decanted. The sludge remaining at the bottom of the reaction vessel was used for pretreatment to test the effectiveness of primary sedimentation of a mixture of raw wastewater with previously generated sludge.

The oily wastewater was first mixed to obtain a homogeneous suspension. In total, 80 L of the homogeneous suspension was mixed with 40 L of generated electrochemical sludge, which was then mixed with air for 30 min using a TetraTec APS 400 pump. Mixing was followed by 30 min of sedimentation, decanting, and sampling.

Subsequently, the oily wastewater was treated in batches using the EF/EC process. First, a set of stainless steel electrodes was placed on the bottom of the reactor tank and the wastewater was treated with 100 A for 15 min, followed by settling and decantation. Then the treatment was continued with iron electrodes for 30 min and the current was 23 A for settling and decanting. The last step included treatment with aluminum electrodes at 20 A for 30 min.

### 3.3. Sample Preparation

### 3.3.1. Determination of the Properties of Raw Oily Wastewater and Treated Water

Electrical conductivity (EC), dissolved oxygen (DO), and pH were determined using a Hanna HI98194 portable multimeter. The minimum detection limit for the parameters EC, DO and pH was 200 mS/cm, 0.01 mg/L.

Chemical oxygen demand (COD) was determined using a NANOCOLOR 500 D spectrophotometer, Macherey Nagel. The NANOCOLOR VARIO C2 thermoblock, Macherey Nagel, was used to digest the samples. The digestion time was 2 h at 148 °C. After digestion, the concentration COD was measured using a NANOCOLOR 500 D spectrophotometer (Macherey Nagel).

To determine the concentration of total hydrocarbons or mineral oils in wastewater samples, the samples were first prepared by the extraction process. To prepare a real sample for analysis, 90 mL of the wastewater sample ($V_{sample}$) and 5 mL of the extraction solvent (n-heptane) first had to be measured out in a beaker, placed in a beaker, and mixed with a magnetic stirrer for 30 min. The sample was then left in the extraction funnel on the stand until the aqueous and organic phases had separated and the aqueous phase could be separated from the organic phase. The organic phase is an extract, the volume of which was measured using a beaker ($V_{extract}$) from which 1 mL of sample was taken, which was then placed in a GC vial. The concentrations of the prepared samples ($c$) were measured using a Nexis Shimadzu GC-2030 instrument. The concentration of mineral oil ($c_{mineral\ oil}$) is calculated according to the following formula:

$$c_{mineral\ oil} = \frac{c \times V_{extract}}{V_{sample}} \tag{1}$$

Analyses were carried out using a Nexis GC-2030 system (Shimadzu, Kyoto, Japan) equipped with a split/splitless injector, autosampler for injecting liquid samples and FID detector. In order to achieve optimal separation, SH-Rxi-5MS column (30 m, 0.25 mm ID, 0.25 μm) was used. The injector unit was set at 290 °C. The oven temperature program was held 1.5 min at 35 °C and raised to 60 °C (5 °C/min); next, the temperature was raised to 315 °C (5 °C/min) and finally held at this temperature for 10 min in order to precondition the column before the next analysis. Nitrogen was used as the carrier gas at constant flow rate of 1.77 mL/min. In total, 1 μL of each sample was injected in the splitless mode. The detection limit (LOD) of the GC method is 0.01 mg/mL.

The analysis of the content of elements in wastewater samples was performed with an instrument from Perkin Elmer DRC ICP-MS. Samples are prepared by placing 1 mL of the wastewater sample and 9 mL of $HNO_3$ in a polypropylene test tube with a volume of

15 mL and mixing gently to achieve homogeneity of the sample. All elemental analysis were carried out by inductively coupled plasma quadrupole mass spectrometry (ICP-MS PerkinElmer SCIEX™ ELAN® DRC-e, Concord, ON, Canada). ICP-MS was used with continuous nebulization. The operating conditions were: Nebulizer Gas flow rates: 0.93 L/min; Auxiliary Gas Flow: 1.2 L/min; Plasma Gas Flow: 14 L/min; Lens Voltage: 8.5 V; ICP RF Power: 1100 W; CeO/Ce = 0.016; Ba++/Ba+ = 0.015. Calibration of the ICP-MS was performed by using of certified standards. For compensation of the possible drift of measurements, internal standards were used. In total, 5 mL of each sample was injected. The detection limit (LOD) of the ICP-MS method is 100 ng/L.

### 3.3.2. Determination of the Properties of Generated Electrochemical Sludge

The sludge sample resulting from the EF/EC treatment was first homogenized and then dried by conventional methods at 105 °C to constant weight in an MRC Mechanical Convection Oven DFO-240N. The samples were ground using a mortar and pestle, dry sieved ($\varnothing$ = 45 μm) and pressed into thick pellets ($\varnothing$ = 2.5 cm) weighing about 2 g. Afterwards, all samples were analyzed using the EDXRF technique, with the excitation source being a Siemens X-ray tube with Mo anode and Mo secondary target in orthogonal geometry. The x-ray tube operated at 45 kV and 35 mA. The samples were irradiated for 1000 s in vacuum and the spectra were collected using a Canberra Si(Li) detector with 3 mm thickness, 30 mm$^2$ active area, 0.025 mm Be window thickness, and a resolution of 170 eV (FWHM) at 5.9 keV. The spectra were analyzed using the IAEA QXAS software and the concentrations of K, Ca, Ti, V, Cr, Mn, Fe, Ni, Cu, Zn, Ga, As, Br, Rb, Sr, Y, Zr, Pb, and Th were determined using the direct comparison of count rates with the IAEA-SL-1 (trace and minor elements in lake sediment) standard reference material.

The moisture content of the generated electrochemical sludge samples was determined by calculating the weight loss after heating the appropriate mass of the sample to 105 °C in an MRC Mechanical Convection Oven DFO-240N dryer to constant weight. The mass loss due to annealing was determined by calculating the weight loss after annealing in the Estherm demiterm Easy9 laboratory furnace (voltage 230 V/50 Hz, 3.0 kW, maximum temperature 1150 °C). Annealing of the specimen was carried out until a constant mass was reached at a temperature of 550 °C.

## 4. Results and Discussion

### 4.1. Characterization of Raw Wastewater Samples

The physicochemical parameters of the oily raw wastewater are listed in Table 2. The oily wastewater is characterized by an odor, a light black color, suspended particles, and a neutral pH. The raw wastewater contains high concentrations of COD (1500–1700 mg/L) and mineral oils (3–130 mg/L). The contents of all other elements are listed in Table 2.

**Table 2.** Physico-chemical parameters of the sample of the oily raw wastewater and the upper permissible limit prescribed by Croatian regulatory body (UPL) for wastewater suitable for discharge into public sewer system (PSS).

| Parameter | Measuring Unit | Oily Wastewater | UPL |
|:---:|:---:|:---:|:---:|
| EC | μS/cm | 521–600 | |
| DO | mg/L | 2.13–4.2 | |
| pH | | 6.89–7.08 | 6.5–9.5 |
| Mineral oil | mg/L | 3–130 | 30 |
| COD | mg/L | 1500–1700 | 700 |
| Al | mg/L | 0.0128–0.0138 | 3 |
| As | mg/L | 0.00017–0.00054 | 0.5 |
| Ca | mg/L | 7.5–12 | |
| Cd | mg/L | 0.0000025–0.00025 | 0.1 |
| Cr | mg/L | 0.00003–0.0022 | 0.5 |

**Table 2.** *Cont.*

| Parameter | Measuring Unit | Oily Wastewater | UPL |
|:---:|:---:|:---:|:---:|
| Cu | mg/L | 0.0000015–0.00068 | 0.5 |
| Fe | mg/L | 0.00003–0.205 | 10 |
| K | mg/L | 0.8–10.5 | |
| Mg | mg/L | 0.5–1400 | |
| Mn | mg/L | 0.00018–0.0325 | 4 |
| Na | mg/L | 2.5–63 | |
| Ni | mg/L | 0.0001–0.0018 | 0.5 |
| Pb | mg/L | 0.00004–0.00012 | 0.5 |
| Si | mg/L | 0.5–1.8 | |
| Sn | mg/L | 0.000027–0.00053 | 2 |
| Zn | mg/L | 0.00075–0.0023 | 2 |

*4.2. Characterization of the Treated Water*

4.2.1. Characterization of Water Samples after Primary Sedimentation of Raw Wastewater and EF/EC Process

A series of experiments tested the effectiveness of pretreatment, primary sedimentation of raw wastewater, in combination with the EF/EC process. The application of pretreatment reduced the concentration of mineral oils (1.7 mg/L), COD (1433 mg/L) by 2.3 and 1.1 times, respectively. In addition, pretreatment had a significant effect on reducing the concentrations of the elements Cr (0.000694 mg/L), Cu (0.000336 mg/L), Sn (0.000012 mg/L) by 3-fold and somewhat less effective by 1.5-fold for As (0.000104 mg/L), Fe (0.16 mg/L), Mn (0.026872 mg/L), Pb (0.000089 mg/L), and Zn (0.002276 mg/L). Primary sedimentation of raw wastewater was found to be effective in removing mineral oil concentration by 57% (Figure 2), and the removal efficiency of COD was much lower at 9.5% (Figure 3).

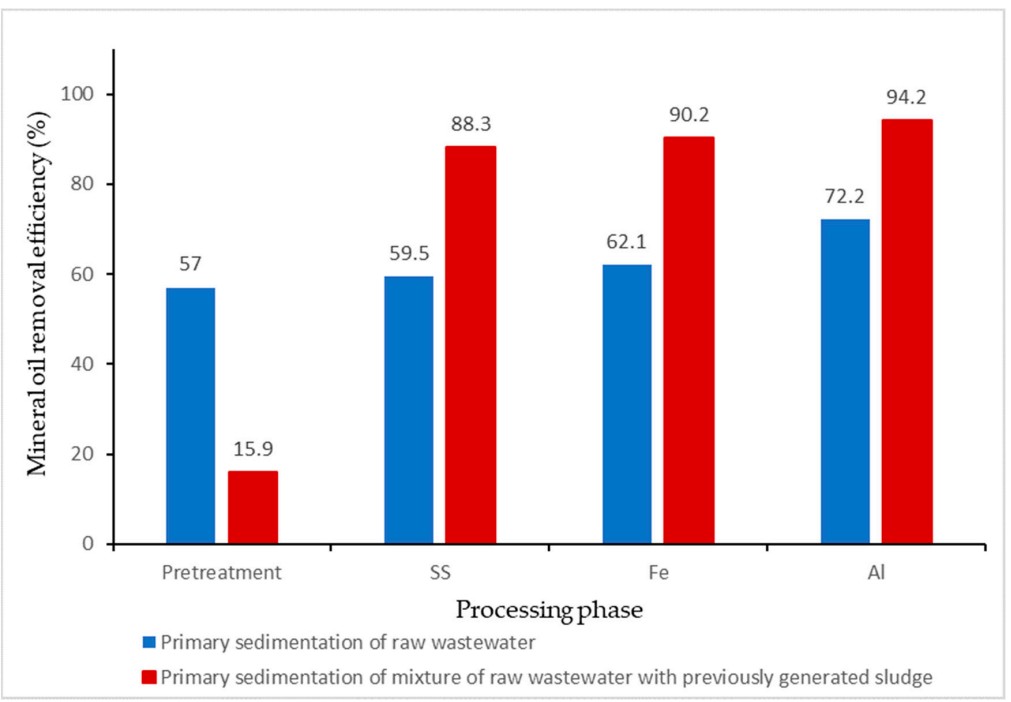

**Figure 2.** Comparison of removal efficiency of mineral oil by primary sedimentation of raw wastewater followed by EF/EC process, during individual treatment stages using stainless steel (SS), iron (Fe) and aluminum (Al) electrodes, and primary sedimentation of a mixture of raw wastewater with previously generated sludge followed by EF/EC process, during individual treatment stages using stainless steel (SS), iron (Fe), and aluminum (Al) electrodes.

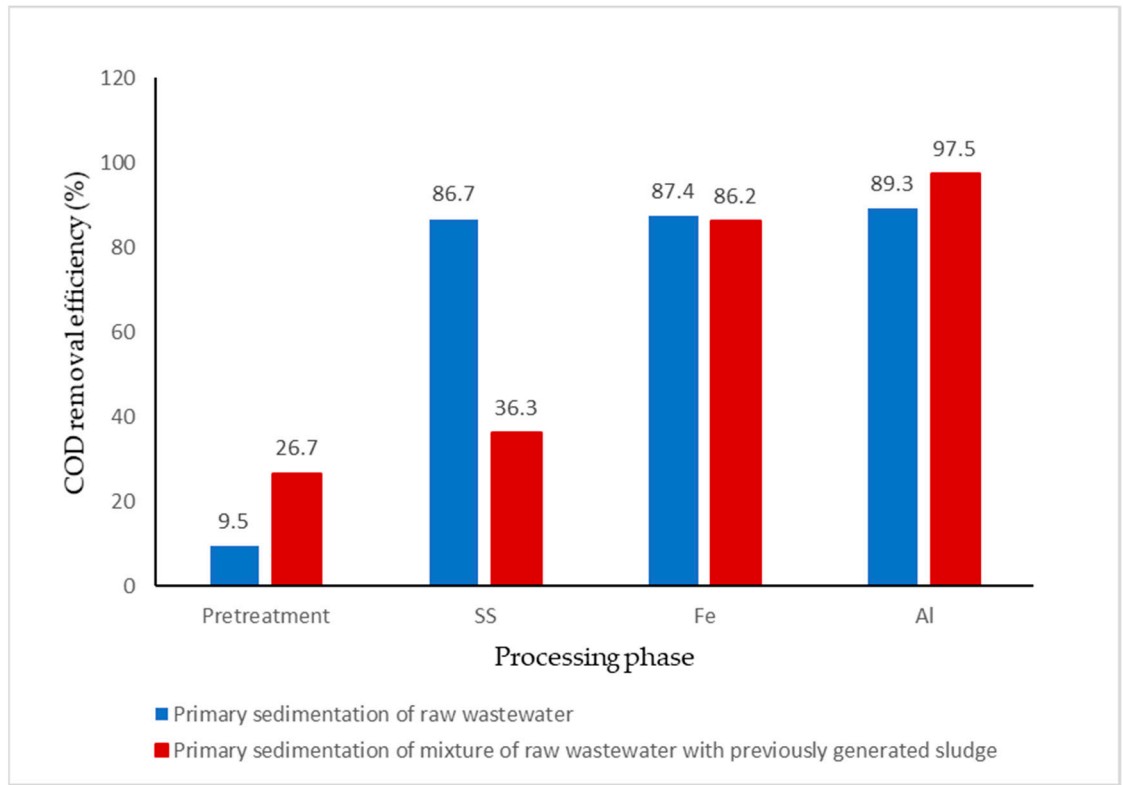

**Figure 3.** Comparison of removal efficiency of COD by primary sedimentation of raw wastewater followed by EF/EC process, during individual treatment stages using stainless steel (SS), iron (Fe), and aluminum (Al) electrodes, and primary sedimentation of a mixture of raw wastewater with previously generated sludge followed by EF/EC process, during individual treatment stages using stainless steel (SS), iron (Fe), and aluminum (Al) electrodes.

After processing EF/EC, the final value of mineral oil concentration was 1.1 mg/L and COD was 170 mg/L, which corresponded to a decrease of 3.6 and 9 times, respectively. The concentrations of individual elements As (0.000025 mg/L), Ca (1.28 mg/L), Cr (0.00035 mg/L), Cd (0.000002 mg/L), Fe (0.06 mg/L), Mn (0.000911 mg/L), Pb (0.000008 mg/L), Sn (0.2 0.000002 mg/L), Zn (0.000458 mg/L) continued to decrease, and Cu and Si were also below the detection limit. The increase in the concentration of Al and Ni in the treated wastewater is a consequence of the use of electrodes made of aluminum and stainless steel, which contains nickel in its composition. The application of primary sedimentation of raw wastewater and EF/EC resulted in a linear increase in the removal efficiency of mineral oils by 72.2% (Figure 2) and COD by 89.3% (Figure 3), and was also effective in the removal of heavy metals. Table 3 shows the values of the chemical parameters determined in the untreated oily wastewater, after the primary sedimentation of the raw wastewater and after the EF /EC process and the final removal efficiency.

4.2.2. Characterization of Water Samples after Primary Sedimentation of Mixture of Raw Wastewater with Previously Generated Electrochemical Sludge and EF/EC Process

In the second series of experiments, primary sedimentation of a mixture of raw oily wastewater with previously generated sludge was combined with the EF/EC process. The addition of the generated sludge reduced the concentration of mineral oil (110.3 mg/L) by 1.2 times and COD (1242 mg/L) by 1.4 times. The application of primary sedimentation of a mixture of raw wastewater and previously generated sludge did not significantly affect the reduction of mineral oils by 15.9% (Figure 2) and COD by 26.7% (Figure 3).

**Table 3.** Chemical parameters determined in untreated oily wastewater, after primary sedimentation of raw wastewater and after EF/EC process, and final removal efficiency.

| Parameter | Measuring Unit | Raw Wastewater | After Pretreatment | After EF/EC Treatment | Removal Efficiency (%) |
|---|---|---|---|---|---|
| Mineral oil | mg/L | 3.96 | 1.7 | 1.1 | 72.19 |
| COD | mg/L | 1583 | 1433 | 170 | 89.26 |
| Al | mg/L | 0.01 | 0.006 | 0.86 | 0 * |
| As | mg/L | 0.000173 | 0.000104 | 0.000025 | 85.26 |
| Ca | mg/L | 7.89 | 7.72 | 1.28 | 83.79 |
| Cd | mg/L | 0.0000026 | 0 | 0.0000002 | 92.99 |
| Cr | mg/L | 0.002272 | 0.000694 | 0.00035 | 84.60 |
| Cu | mg/L | 0.000675 | 0.000336 | 0 | 100 |
| Fe | mg/L | 0.21 | 0.16 | 0.06 | 71.76 |
| K | mg/L | 0.86 | 0.83 | 0.66 | 23.23 |
| Mg | mg/L | 0.52 | 0.50 | 0.16 | 69.04 |
| Mn | mg/L | 0.032606 | 0.026872 | 0.000911 | 97.21 |
| Na | mg/L | 2.92 | 2.86 | 2.49 | 14.79 |
| Ni | mg/L | 0.000109 | 0 | 0.002618 | 0 * |
| Pb | mg/L | 0.000113 | 0.000089 | 0.0000077 | 93.17 |
| Si | mg/L | 0.81 | 0.79 | 0 | 100 |
| Sn | mg/L | 0.000027 | 0.000012 | 0.0000002 | 99.27 |
| Zn | mg/L | 0.002704 | 0.002276 | 0.000458 | 83.07 |

* Higher concentration in treated water compared to raw wastewater.

The following application of EF/EC resulted in a significant final treatment efficiency with effluent concentration of mineral oil of 7.6 mg/L (17.25 times less) and COD of 43 mg/L (39.4 times less), respectively. The concentrations of other elements As (0.000186 mg/L), Ca (3.94 mg/L), Cr (0.000009 mg/L), Na (0.72 mg/L), Ni (0.000008 mg/L) further decreased, and Cd, Cu K, Pb, Si, Sn, and Zn were completely removed. The increase in the concentration of Al, Fe, and Mn in the treated wastewater is a consequence of the use of electrodes made of iron, aluminum, and stainless steel, which is contained within its composition. The combination of primary sedimentation of the mixture of raw wastewater and previously generated electrochemical sludge followed by EF/EC process showed a very high mineral oil removal efficiency of 94.2% (Figure 2) and COD of 97.5% (Figure 3). Table 4 shows the values of the chemical parameters determined in the untreated oily wastewater, after the primary sedimentation of the mixture wastewater with previously generated sludge and after the EF /EC process and the final removal efficiency.

**Table 4.** Chemical parameters determined in untreated oily wastewater, after primary sedimentation of mixture of raw wastewater with previously generated sludge and after EF/EC process, and final removal efficiency.

| Parameter | Measumerent Unit | Raw Wastewater | After Pretreatment | After EF/EC Treatment | Removal Efficiency (%) |
|---|---|---|---|---|---|
| Mineral oil | mg/L | 131.1 | 110.3 | 7.6 | 94.20 |
| COD | mg/L | 1694 | 1242 | 43 | 97.46 |
| Al | mg/L | 0.013783 | 0.014503 | 0.000034 | 0 * |
| As | mg/L | 0.000537 | 0.000342 | 0.000189 | 65.36 |
| Ca | mg/L | 11.68 | 0 | 3.94 | 66.27 |
| Cd | mg/L | 0.000246 | 0.00044 | 0 | 100 |
| Cr | mg/L | 0.000032 | 0.000011 | 0.000009 | 71.03 |
| Cu | mg/L | 0.000002 | 0 | 0 | 100 |
| Fe | mg/L | 0.000033 | 0.000035 | 0.000097 | 0 * |
| K | mg/L | 10.53 | 2.82 | 0 | 100 |
| Mg | mg/L | 1350 | 24,326 | 2225 | 0 * |
| Mn | mg/L | 0.000182 | 0.000061 | 7.41 0.00007 | 0 * |
| Na | mg/L | 62.77 | 9.21 | 0.72 | 98.85 |
| Ni | mg/L | 0.001799 | 0.000029 | 0.000008 | 99.40 |
| Pb | mg/L | 0.000042 | 0.000014 | <0.1 | 100 |
| Si | mg/L | 1.737 | 1.544 | <0.1 | 100 |
| Sn | mg/L | 0.000527 | 0.000524 | <0.1 | 100 |
| Zn | mg/L | 0.000793 | 0.000005 | <0.1 | 100 |

* Higher concentration in treated water compared to raw wastewater.

Figure 2 shows a comparison of the efficiency of mineral oil removal in primary sedimentation of raw wastewater and primary sedimentation of a mixture of raw wastewater and previously generated electrochemical sludge, both followed by EF/EC process using stainless steel (SS), iron (Fe), and aluminum (Al) electrodes. Pretreatment of primary sedimentation of raw wastewater achieved 57% efficiency compared to primary sedimentation of the mixture of raw wastewater with previously generated sludge of 16% for mineral oil. Primary sedimentation of a mixture of raw wastewater with previously generated sludge achieved a higher final efficiency of 94%, whereas primary sedimentation of raw wastewater achieved an efficiency of only 72% for mineral oil. With a higher load of oily wastewater and a mineral oil concentration of 131.1 mg/L, the technological sequence with pretreatment of raw wastewater and primary sedimentation of the mixture of raw wastewater with previously generated sludge proved to be more effective and achieved a mineral oil removal efficiency of over 90%.

Figure 3 shows a comparison of the effectiveness of removal of COD by primary sedimentation of raw wastewater and primary sedimentation of a mixture of raw wastewater with previously generated sludge followed by EF/EC process during certain treatment stages using stainless steel (SS), iron (Fe), and aluminum (Al) electrodes. For COD removal, pretreatment of raw wastewater using primary sedimentation achieved an efficiency of 9.5%, whereas mixing of raw wastewater and primary sedimentation of the mixture of raw wastewater with previously generated sludge achieved an efficiency of 26.7%. A higher efficiency of final COD removal was achieved by primary sedimentation of the mixture of raw wastewater with previously generated sludge with 98%, whereas primary sedimentation of raw wastewater achieved an efficiency of 89%. This proves the reactivity of the electrochemically generated sludge. Moreover, it proven to be effective already during pretreatment, which is later reflected in a higher efficiency after the whole treatment process (pretreatment, SS, Fe, Al).

Figure 4 shows a comparison of the final removal efficiencies of mineral oil, COD, As, Ca, Cd, Cr, Cu, Ni, Pb, Sn, Zn using primary sedimentation of raw wastewater followed by EF/EC process, and primary sedimentation of a mixture of raw wastewater with previously generated sludge followed by the EF/EC process.

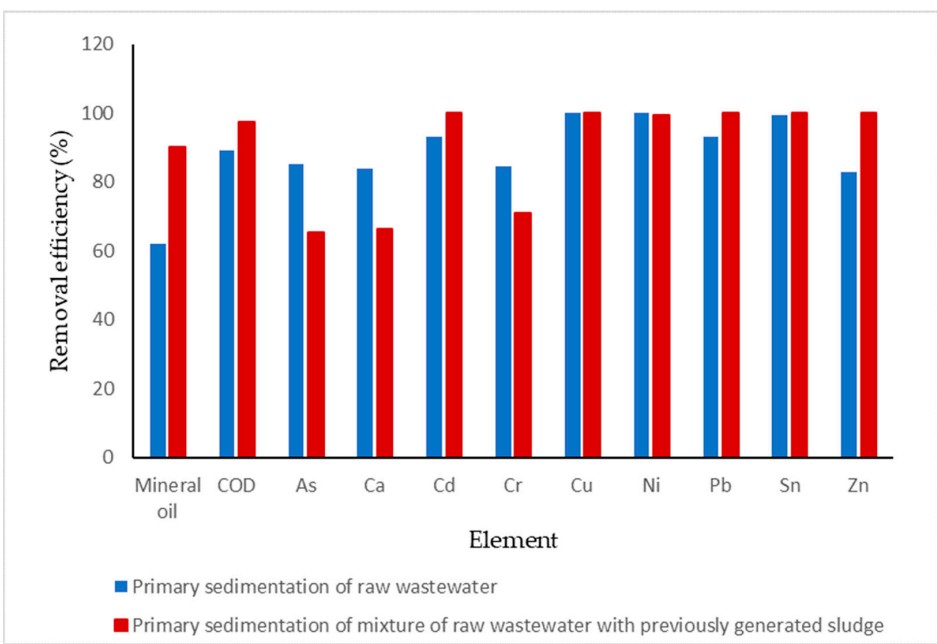

**Figure 4.** Comparison of removal efficiency of mineral oil, COD, As, Ca, Cd, Cr, Cu, Ni, Pb, Sn, Zn by primary sedimentation of raw wastewater followed by EF/EC process, and primary sedimentation of a mixture of raw wastewater with previously generated followed by EF/EC process.

For most heavy metals, after the whole treatment process (pretreatment, SS, Fe, Al), a higher final treatment efficiency was obtained when primary sedimentation of a mixture of raw wastewater with previously generated electrochemical sludge was used.

*4.3. Analysis of Electrochemical Sludge Generated during EF/EC Treatment of Oily Wastewater*

4.3.1. Determination of the Content of Elements in Generated Electrochemical Sludge

Electrodes made of stainless steel, iron, and aluminum were used in the treatment of oily wastewater with the EF/EC process. Table 5 shows the content of each element in the generated sludge. The higher content of iron (7.73%) is the result of the use of iron electrodes during the treatment process of oily wastewater. The content of elements chromium (269.7 ppm), nickel (148.3 ppm), copper (85 ppm), zinc (441.3 ppm), and manganese (1314 ppm) in the separated sludge is the result of using stainless steel and iron electrodes.

**Table 5.** Mass concentrations and standard deviations (SD) of elements in electrochemical sludge generated during the treatment of oily wastewater.

| Element | Measumerent Unit | Solid Sample |
|---|---|---|
| K | % | $0.18 \pm 0.03$ |
| Ca | % | $1.62 \pm 0.09$ |
| Fe | % | $7.73 \pm 0.20$ |
| Ti | ppm | $175 \pm 24$ |
| V | ppm | $10.1 \pm 1.8$ |
| Cr | ppm | $269.7 \pm 26.8$ |
| Mn | ppm | $1314 \pm 61$ |
| Ni | ppm | $148.3 \pm 28.6$ |
| Cu | ppm | $85 \pm 17$ |
| Zn | ppm | $441.3 \pm 20.0$ |
| Ga | ppm | $33.3 \pm 7.2$ |
| As | ppm | $6.4 \pm 0.7$ |
| Br | ppm | $4 \pm 0.3$ |
| Rb | ppm | $8.3 \pm 0.8$ |
| Sr | ppm | $104 \pm 56$ |
| Y | ppm | $4.3 \pm 0.3$ |
| Zr | ppm | $63 \pm 4$ |
| Pb | ppm | $9.3 \pm 1.9$ |
| Th | ppm | <0.89 |

4.3.2. Determination of Moisture Content and Loss on Ignition of Generated Electrochemical Sludge

Treatment of oily wastewater using the EF/EC process in combination with pretreatment results in generating the sludge whose moisture content and mass loss are determined by annealing. The moisture content in the sludge sample was 2.4%. The loss on ignition represents the percentage of crystalline water and organic matter and was 26.1%. The electrochemical sludge is characterized by a relatively high pH of 9.45.

**5. Conclusions**

The results of this study show that for oily wastewater of mineral origin, pretreatment by primary sedimentation of the raw wastewater was found to be effective in removing mineral oil and COD. The initial concentration of mineral oil was 3.96 mg/L, whereas COD was 1583 mg/L. With this type of pretreatment, a mineral oil concentration was 1.7 mg/L (removal efficiency of 57%) and COD 1433 mg/L (removal efficiency of 10%), whereas after the whole treatment process (pretreatment followed by EF/EC), a mineral oil concentration was reduced to 1.1 mg/L (removal efficiency of 72%) and COD concentration was reduced to 170 mg/L (removal efficiency of 89%). Moreover, the removal efficiency of the elements As, Ca, Cr, and Zn was over 85%, Cd and Pb were over 90%, and Cu and Ni were almost completely removed.

In the second series of experiments, the efficiency and reactivity of the electrochemically generated sludge was demonstrated by mixing with raw wastewater as pretreatment prior to primary sedimentation pretreatment and before the final electrochemical EF/EC treatment process. The initial concentration of mineral oil was 131.1 mg/L, while at COD it was 1694 mg/L. The primary sedimentation pretreatment, i.e., mixing the raw wastewater with the previously generated electrochemical sludge, did not achieve significant efficiency in mineral oil removal 16% (110.3 mg/L) compared to the primary sedimentation of raw wastewater, whereas a slightly higher efficiency of 27% was achieved for COD (1242 mg/L). The slightly lower efficiency of this pretreatment is likely due to the fact that some of the organic loading was introduced by the addition of the previously generated electrochemical sludge. However, the addition of sludge resulted in higher final efficiencies (pretreatment followed by EF/EC) for mineral oil 95% (7.6 mg/L) and COD 98% (43 mg/L) and for elements such as Cd, Cu, Ni, Pb, Sn, and Zn 100%. Pretreatment with mixing the generated electrochemical sludge with raw wastewater prior to the primary sedimentation pretreatment process was found to be effective when higher input loads are present, which is very important due to the possibility of different concentrations of input loads in raw oily wastewater, dependent on its origin. As part of future research, it is proposed to test other indicators of water quality, including BOD and TOC.

**Author Contributions:** Conceptualization, D.V. and T.B.; methodology, M.D., D.V. and T.B.; validation, D.V. and T.B.; formal analysis, M.D.; investigation, M.D., D.V. and H.P.; resources, M.D. and D.V.; data curation, M.D., D.V. and H.P.; writing—original draft preparation, M.D. and D.V.; writing—review and editing, D.V. and T.B.; visualization, M.D.; supervision, D.V. and T.B.; project administration, D.V. All authors have read and agreed to the published version of the manuscript.

**Funding:** This research was funded by the Croatian Science Foundation under the project "IP-2019-04-1169-Use of treated oily wastewater and sewage sludge in brick industry-production of innovative brick products in the scope of circular economy".

**Institutional Review Board Statement:** Not applicable.

**Informed Consent Statement:** Not applicable.

**Data Availability Statement:** Not applicable.

**Acknowledgments:** This work has been fully supported by the Croatian Science Foundation under the project "IP-2019-04-1169-Use of treated oily wastewater and sewage sludge in brick industry-production of innovative brick products in the scope of circular economy".

**Conflicts of Interest:** The authors declare no conflict of interest. The funders had no role in the design of the study; in the collection, analyses, or interpretation of data; in the writing of the manuscript; or in the decision to publish the results.

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
