# Peer review of "The Influence of Pretreatment on the Efficiency of Electrochemical Processes in Oily Wastewater Treatment"

_water, doi:10.3390/w14192976_

Round 1

Reviewer 1 Report

The overall manuscript is well written and clear results have been shown. However, introduction section requires few changes to strengthen the reference studies cases. 
E.g., Anaerobic digestion references could be added as wastewater is used to treat and produce biogas. 
https://www.sciencedirect.com/science/article/abs/pii/S0959652618327343

Similarly for MBR technology and its application, refer to the following
https://pubs.rsc.org/en/content/articlehtml/2018/ra/c8ra03810d

Overall the manuscript it of great interest. 

Reviewer 2 Report

Dear authors, your manuscript is very good, clear and you made a very well elaborated analysis on the effect of pretreatment on waste treatment. The suggestion that I can make is that try to compare their results with what is obtained with other processes such as adsorption to enhance their results.

Reviewer 3 Report

The manuscript by Druskovic et al. reports about potential pretreatment of mineral oil wastewater. Generally, the topic is interesting and fits within the scope of the chosen Journal Waters. Unfortunately, I have serious doubts that this submitted manuscript can be accepted for publication. There are too many obstacles to be revised. In particular, the introduction is by far too long and too general. The introduction includes three subchapters. In my opinion this is by far too long for a research paper. Reading carefully through all three chapters I am not convinced that they are necessary. The introduction might be drastically shortened to half of the length current of the introduction. Especially chapter 1 is written too general. Instead of introduction each conventional treatment methods, the authors should provide a more detailed overview using Table 1. The information of Table 1 in the submitted manuscript is only a very poor summary the introduction without any additional content. The aim of the study is not clearly indicated and thereby it remains unclear why the main focus is turned on metal analyses of the treated sludge. There are several missing information in the material and method section. For example, the COD analysis is incomplete. The GC method is not described as well as the ICP-MS method. This is unacceptable for a scientific paper. The result and discussion section shows several layout problems with Tables presented. In addition, the is no consistent use of units between the different Tables making a comparison between them difficult to unexperienced readers. Figures have misleading axes labeling. The quality of the Figures is very poor, too. As I have seen some other publications of the authors, I know that they can write very high-quality papers. The submitted paper presented here is not representing their common scientific quality. In order to maintain the high quality of the Journal Waters, I have to recommend to reject the submitted manuscript. Please find some general information below supporting my decision: 

Line 36/59: This part of the introduction seems very general and too long. Please shorten this part. 

Line 77: What do you mean with “complex control system”?

Line 87/88: This statement is too strong and by the way not true. Please “soften” your statement and make it more relative.

Line 92: EPS are well known as important function in forming bioflocs and/or biofilms. The role as flocculant requires here at least a strong reference and more detailed description. 

Line 99/100: This is just a relative removal efficiency. Please report in terms of absolute removal efficiency for better comparison. 

Line 132: Table 1 shows only disadvantages of the conventional techniques. However, they were developed due to some relevant advantages. I recommend including advantages vs disadvantages in Table 1. 

Line 277: The aim of the study is missing at the end of the introduction. 

Line 283: What was analysed to determine which properties?

Line 341/344: The method description is incomplete. What did you do after the digestion? Commonly the digestion for COD is done at 148°C for 2 h. How can you ensure that your digestion was complete after 160°C? This is not a standard method!

Line 345/355: Method is misleading. How is the organic phase extracted? And which method was used for GC analyses? The complete GC method is missing!

Line 356/359: The ICP-MS method is completely missing!

Line 373: This table does not belong to the M&M section, isn’t it?

Line 383: Should this be table 2 or 3? Layout and/or format problem?

Line 394/410: As I understood the initial mineral oil concentration of your water was 3.96 mg/L. According to table 2 the UPL requires 30 mg/L for discharge. Why do you need to treat your water to decrease the mineral oil content? There is no reason. Of course, drastic reduction of COD and metal concentration is urgently required, but as a reader I did not understand that it was your main aim!

Line 412: I recommend reporting understandable concentrations. I have serious doubts that your instrument can measure nano grams. Certainly, you enriched your samples and measure higher concentrations. Please use consistent units. And why did you measure all these metal? Why is Ca, Mg, K, Fe and others important. Do they real pose an environmental risk?

Line 432: Why is the concentration of Al and Fe after EF/EC treatment drastically increased? 

Line 447: Misleading figure information. The declaration of the Y axes is intending that mineral oil is removed, but on the X axes information on metal concentration is delivered. Thus, it is the removal of metal of oily sludge that is removed. Any the Figure quality is very bad. This goes for all Figures!

Line 465: Similar problem as for Figure 2!

Line 487: Headline of table is missing and mixture of units! Here you report ppm while in the upper Tables mg/L or other mass concentration were used!

Round 2

Reviewer 3 Report

The revised manuscript has been improved by the authors. However, I still have serious doubts about impact of the scientific outcome. The authors show that water from traffic areas can be treated with EF/EC achieving high removal efficiency. Considering the composition of the oily wastewater, I am wondering why EF/EC is necessary? Since the BOD or at least TOC is missing almost no relation to COD is possible. It remains in speculation whether simple ozonation might be a more suitable treatment. Nevertheless, I understand that the authors want to stress the fact that they wanted to investigate the influence of two different pretreatments. In that context, the authors may please justify in the discussion why they have chosen this specific effluent water. 

The introduction was drastically shortened. This is very welcome. However, find some relevant comments concerning the introduction and other parts of yours revised manuscript below:

Line 21 to 25: The abstract sounds again very general. Especially, this sentence must be revised. It is too long and confusing. Please shorten it by using a full stop in between. 

Line 83: This shall be chapter 2, I guess?

Line 84 to 98 is too general and reads rather a common literature review than a straightforward introduction to your scientific paper. Please prove whether this complete part can be shortened or deleted. Overall, your introduction is very long. 

Line 262: Are you sure that you injected 1 µL?

Line 273: The injection volume of your sample is missing. Please include the limit of detection and determination of your Gc and ICP-MS method.

Line 298 & 299: What is it an “unpleasant odor”, or an “intense odor” or an unpleasant intense odor? Please revise

Line 300: Why do you present a concentration range in table 2, but in line 300 only a fixed concentration is mentioned? Revise

Line 308/310: Table 2 – I recommend inserting an extra column for the unit. This will help the reader to better understand reported values. This should be done with all tables in your manuscript. And again another remark, if the concentrations of your heavy metals is so very low you may discuss that also in your paper. I mean is there any heavy metal having a critical concentration? Thus, even if your relative removal efficiencies are high, they might not mean that much. This is an important fact to be discussed in your discussion. 

Line 334 and 357, i.e. Table 3 & 4: I recommend to insert another column with the relative removal efficiency after EF/EC treatment. By doing so, your relative removal efficiencies in the main text are justified. 

Line 372 and 391, i.e. Figure 2 & 3: Still the quality of your figure is very low. Even if you use excel, please produce better quality. Eliminate the frame. Be aware that in English language on comma is used for numbers but a point. The resolution is poor. Save your graphs a picture with high resolution. 

Line 415: The headline of your table 5 is below the table. Please revise. If you have determined the concentration in terms of mg/kg you should also report it in this way. Please try to be as clear as possible with the reader. Again, I recommend inserting another column for the unit. 

Finally, please check the manuscript for tipping errors. 

Round 3

Reviewer 3 Report

The manuscript has been well improved and is now acceptable for publication.